# Trajectories of freshwater microbial genomics and greenhouse gas saturation upon glacial retreat

Jing Wei[1], Laurent Fontaine[1], Nicolas Valiente [1,2], Peter Dörsch[3], Dag O. Hessen[1] & Alexander Eiler [1] ✉

Due to climate warming, ice sheets around the world are losing mass, contributing to changes across terrestrial landscapes on decadal time spans. However, landscape repercussions on climate are poorly constrained mostly due to limited knowledge on microbial responses to deglaciation. Here, we reveal the genomic succession from chemolithotrophy to photo- and heterotrophy and increases in methane supersaturation in freshwater lakes upon glacial retreat. Arctic lakes at Svalbard also revealed strong microbial signatures form nutrient fertilization by birds. Although methanotrophs were present and increased along lake chronosequences, methane consumption rates were low even in supersaturated systems. Nitrous oxide oversaturation and genomic information suggest active nitrogen cycling across the entire deglaciated landscape, and in the high Arctic, increasing bird populations serve as major modulators at many sites. Our findings show diverse microbial succession patterns, and trajectories in carbon and nitrogen cycle processes representing a positive feedback loop of deglaciation on climate warming.

The Ice Sheets in the Arctic and Antarctic as well as glaciers at low latitudes have been losing mass for several decades at an accelerating rate[1–3] associated with a steady increase of approximately 2 °C in air temperature over the past 30 years[4] with the Arctic warming four times faster than the global average in recent decades; in some high-Arctic areas even exceeding 2 °C per decade[5,6]. Deglaciation is the primary contributor to twenty-first century sea level rise[7], with important feedbacks to climate by reduced albedo[8] and as a major modulator of terrestrial and aquatic carbon and nutrient cycles[9–11]. The latter is still not well quantified, including atmospheric greenhouse gas (GHG) release and uptake, as we do not understand microbial responses across landscapes following glacier retreat.

Deglaciated landscapes undergo rapid geomorphic change as sedimentological, hydrological and aeolian processes begin to alter the landscape. In high latitudes (e.g. Svalbard), deglaciation is most significant along the coast and many marine-terminating glaciers are

predicted to retreat further and thus calve on land in the near future[12]. Landscape changes following glacial retreat and permafrost melt lead to the disappearance of the 'cryosphere cap' that prevents gas leaking from large reservoirs of geologic $CH_4$ in coal beds, natural gas deposits or gas hydrates[13–15]. Furthermore, thawing permafrost is thought to mobilize permafrost carbon stores[13,16] and could thus release vast amounts of GHG to the atmosphere under continued anthropogenic warming[17,18].

These landscape changes[19,20] are paralleled by primary ecological succession[21] and greening[22] with feedbacks on GHG emissions. Elevated temperatures and retreats of glaciers and permafrost cause a subsequent establishment of vegetation. There have been multiple studies showing how plant communities undergo a gradual increase in structural complexity, biomass, species diversity and ecosystem interaction[23,24], yet few studies have addressed the impact of these changes on aquatic systems at a catchment level. The physical

[1]Department of Biosciences and Centre for Biogeochemistry in the Anthropocene, University of Oslo, 0316 Oslo, Norway. [2]Division of Terrestrial Ecosystem Research, Center of Microbiology and Environmental Systems Science, University of Vienna, 1030 Vienna, Austria. [3]Faculty of Environmental Sciences and Natural Resource Management, Norwegian University of Life Sciences, 1432 Ås, Norway. ✉e-mail: alexander.eiler@ibv.uio.no

conditions of proglacial aquatic systems are likely to be affected by large landscape shifts, with subsequent repercussion on biodiversity and ecological functioning[25]. Clear successional patterns of macroinvertebrates[26,27] and microorganisms[28–31] in lakes and rivers have been described. However, how the functional succession of microbes during lake aging (or lake succession) affects lakes as sinks or sources of GHG, remains poorly understood.

On short term, glacier retreat may increase turbidity in proglacial lakes and coastal marine waters affecting environmental factors such as temperature[32] and light availability in the water column[33], causing unfavorable conditions for phototrophic primary producers. Proglacial lakes may thus be dominated by organoheterotrophic and lithoautotrophic microbes[34,35]. Glaciers may also supply downstream aquatic systems with nutrients and organic C[36,37]. As the contribution of glacier meltwater to aquatic systems diminishes, this leads to clear water and increased primary productivity. In addition, greening in the thawing catchment contributes allochthonous organic matter to the aquatic systems, thus boosting heterotrophic microbial activity and promoting their role as GHG conduits to the atmosphere.

As a side-effect of warming in the Arctic, there has been an extensive growth of breeding populations of geese, providing nutrients to freshwaters[38,39], and thus stimulating algal and microbial production even further[40,41]. Typically, these populations of geese proliferate at the sites near the sea where vegetation is most developed. These changes can be expected to drive a complex microbial succession from chemolithotrophs to phototrophs and heterotrophs, with consequences for the taxonomic and functional dynamics of microbially mediated carbon and nutrient cycling, including the turnover of GHGs, across the deglaciated landscapes.

Here, we present results from a field study of 31 lakes in the proximity of rapidly receding glaciers in Kongsfjorden and Longyearbyen (high Arctic Svalbard), and use a corresponding gradient from a glacier front at an alpine site in central Norway for comparison. Our study design was based on chronosequences which substitute space for time and thus allow to study ecological succession at decadal time-scales[21,42]. Samples were obtained along five distinct chronosequences in five individual catchments defined as a series of lakes of different ages formed due to glacial retreat (see map in Supplementary Fig. 1). Most sites are remote and largely unaffected by direct anthropogenic disturbance, highlighting their sentinel role in the landscape[43]. We studied the effects of glacial retreat on microbial (bacteria and eukarya) diversity and estimate potential functional consequences using metagenomics in relation to GHG saturation. Since many high Arctic sites have been heavily impacted by growing geese populations over the recent decades[38,39], we also assessed bird impacts on the microbial communities.

We hypothesize that the greater the distance from glaciers, which is associated with a higher nutrient concentrations, the higher the microbial diversity, reflecting outcomes of ecological succession[44,45] and the productivity-biodiversity relationship[46,47]. Furthermore, we test the hypothesis that functional diversity assessed by metagenomics follows highly reproducible successional patterns irrespective of biogeographic region (alpine vs. arctic Norway) but is modulated by birds and thawing organic matter. Together, this leads to lake specific GHG emissions, as reflected in $CH_4$, $CO_2$, and nitrous oxide ($N_2O$) oversaturation.

## Results and discussion
### GHG saturation varies non-linearly with lake properties
Along the chronosequences, $CO_2$ saturation revealed net heterotrophy of most studied freshwater systems at the time of sampling, with the exception of 7 systems (Fig. 1A). Average $CO_2$ saturation showed no clear trends across the chronosequences with $285 \pm 187\%$ in recently formed lakes (defined as lakes closer than 550 meters from the glacier front) to $278 \pm 473\%$ in the oldest sampled lakes (defined as lakes more

than 3300 meters from the glacier front), but was tightly linked to concentrations of dissolved organic carbon and total nitrogen concentrations as revealed by generalized additive models (Supplementary Fig. 2). $CH_4$ saturation increased significantly from $181 \pm 98$ to $28,108 \pm 21254\%$ along the transect, representing an increase in $CH_4$ oversaturation of 15529% across the deglaciated landscape. The large variations of $CH_4$ saturation were tightly associated with distance to glacier (a proxy for the age of the lakes; $p_{gl\_dist} = 0.041$, $F_{gl\_dist} = 1.283$) and dissolved organic carbon ($p_{DOC} = 0.002$, $F_{DOC} = 5.259$; $p_{TN} < 0.001$; $F_{TN} = 34.26$) as revealed by generalized additive models (GAMs; Fig. 1B, D for $CH_4$). $N_2O$ saturation showed less variation, and model predictions were generally insignificant ($p$-values > 0.1). The importance of nutrients for $CH_4$ production and concentrations has been shown previously for water and sediments[48–51] as well as for $CH_4$ emissions to the atmosphere[50,52]. However, our GAMs suggest that these relationships are not strictly monotonic over the nutrient gradients, i.e. either positive, negative or neutral relationships exist, challenging the outcomes of previous studies. This is supported by comparing the accuracies of GAMs ($R^2 = 0.952$, GCV = 3399.6, deviance explained = 98% for $CH_4$; $R^2 = 0.484$, GCV = 4.56, deviance explained = 65.9% for $CO_2$) and GLMs ($R^2 = 0.355$, GCV = 20434, deviance explained = 40.9% for $CH_4$; $R^2 = 0.105$, GCV = 5.71, deviance explained = 17.9% for $CO_2$).

Non-monotonic relationships such as between GHG concentrations and environmental variables with both increasing and decreasing sectors suggest that sudden changes can occur when an environmental factor reaches a threshold. Such rapid shifts apparently occur at around 5 and 11 mg DOC $L^{-1}$ and 0.25 mg total nitrogen (TN) $L^{-1}$ for $CH_4$ saturation (Fig. 1C, D). Likewise, thresholds in $CO_2$ concentrations were observed at 0.25 mg TN $L^{-1}$ and 5 and 11 mg DOC $L^{-1}$ across the deglaciated chronosequences (Supplementary Fig. 2). Comparison of these data, together with previous results[53–55], emphasizes the high variability in $CH_4$ and $CO_2$ concentrations of lake systems in the arctic landscape. This variability in gas composition (e.g. $CH_4$:$CO_2$ ratios) and GHG emissions in these shallow and well oxygenated systems are likely driven by in-lake processes in sediments determined by nutrient input from birds (reflected in TN concentrations) as well as dissolved organic carbon from increasing vegetation and thawing permafrost soils[56].

### Microbial succession along the chronosequences
As expected, increases in GHG saturation were associated with increases in total bacterial cell numbers along the chronosequences. Starting with an average of $9.0 \times 10^4$ cells $ml^{-1}$ in lakes closer than 550 meters to the glacier front, cell numbers increased to $2.4 \times 10^5$ cells $ml^{-1}$ in aged lakes more than 3300 meters from the glacier front. Using amplicon sequencing[57], we explored if this doubling in microbial cell numbers was associated with changes in microbial diversity. Using a parallel approach for bacteria and eukaryotes, we identified 6719 unique bacterial and 3483 unique eukaryotic sequences (whereof 112 were Metazoans), with maxima in abundance-based coverage (ACE) ranging from 93 (SV015; Brøggerdalen Glacier) to 838 (SV013; Permafrost pond, Brøggerdalen) and amplicon sequence variant (ASV) richness for bacteria from 50 (SV001; Hotelnesset, Longyearbyen) to 593 (FI011; Finse) (for details on sequencing statistics and data processing see Supplementary Tables 1 and 2). Attempts to amplify archaeal amplicons failed for most samples most likely due to low presence of Archaea, as indicated by shotgun metagenomic data (relative contribution of archaeal reads was on average 0.13%). Bacterial ASV richness increased with distance to the glacier, cell numbers and nutrient status of the lake system, and was associated with bird impact, while eukaryotic richness was associated with conductivity (negatively), $CO_2$ (negatively) and $CH_4$ (positively) concentration/saturation (Fig. 2A). A low diversity in water bodies close to glaciers has been suggested to be due to habitat conditions (i.e. low temperatures, absence of soils, low impact from birds, presence of silts and low nutrient concentrations), less diverse sources of microorganism input

(i.e., aside from the glacier itself), or a combination of both[29]. Inferred differences in the main associations, as revealed by PLS models, suggest a low degree of coupling in alpha diversity between Eukaryotes and Bacteria across the deglaciated chronosequences.

Contrary to the weak relations found between bacterial and eukaryotic richness (Spearman correlation $R = 0.38$, $p = 0.0416$), we found highly overlapping patterns in bacterial and eukaryotic beta diversity (procrustes $R = 0.67$, $p < 0.001$) in the four chronosequences at Svalbard with the Finse samples forming a separated cluster (NMDS plots; Supplementary Fig. 3). Potential explanations for a coupling across the deglaciated landscape are inferred similarities in environmental drivers (Fig. 2B), system connectivity (downstream sites will receive cells from upstream sites) and co-occurrences (potential interactions) across the two domains (as determined by maximal information-based nonparametric exploration[58]; see Supplementary Table 3). While different dispersal sources and rates[59,60] in connection with disparate energetics and generation times[61,62] may lead to a decoupling, as indicated by unrelated alpha diversity estimates of bacteria and eukaryotes.

The overall community patterns were complemented by taxonomic shifts where subgroups associated with lakes with proximity to glaciers decreased while others increased with distance (Supplementary Fig. 4 heatmap of ASVs and for MAGs dynamics see below). As revealed by generalized linear latent variable models, eukaryotes such as *Dinoflagellata* and *Chlorophyta* increased with distance to the

glaciers while *Fungi* and *Chryptophyta* were more common close to the glaciers indicating a succession from heterotrophic fungi to photosynthetic algae (for more detail on the gllvm using eukaryotic ASV dynamics see Supplementary Fig. 5). There was also a shift toward *Cyanobacteria* of the genera *Pseudanabaena* and *Nostoc* in lake systems with bird impact and high nitrogen concentrations (for more detail on the gllvm using bacterial ASV dynamics see Supplementary Fig. 6). In these later successional stages of the lakes *Daphnia* spp. were often observed with *Lepidurus arcticus*, a species of tadpole shrimp, preying on *Daphnia*, both widespread inhabitants of high-Arctic systems devoid of fish[63], corresponding to an increase in one trophic level. These significant changes in bacterial and eukaryotic communities could be observed on top of variability caused by factors such as freezing and thawing, amount of snow and wind across the glacial forefields[42]. As such, these spatial trends in the microbial community emphasize the occurrence of microbial succession and an increase in trophic levels associated with glacial retreat.

### Functional succession along the chronosequences

In addition to succession in microbial taxonomic diversity, we expected the distribution of microbial traits to change. We focused on metabolic traits that are expected to drive C, N and S cycling, and thus the exchange of GHG across the atmosphere-terrestrial interface. To obtain spatial resolution functional trait data, we applied a shotgun metagenomics approach and inferred functional changes encoded in

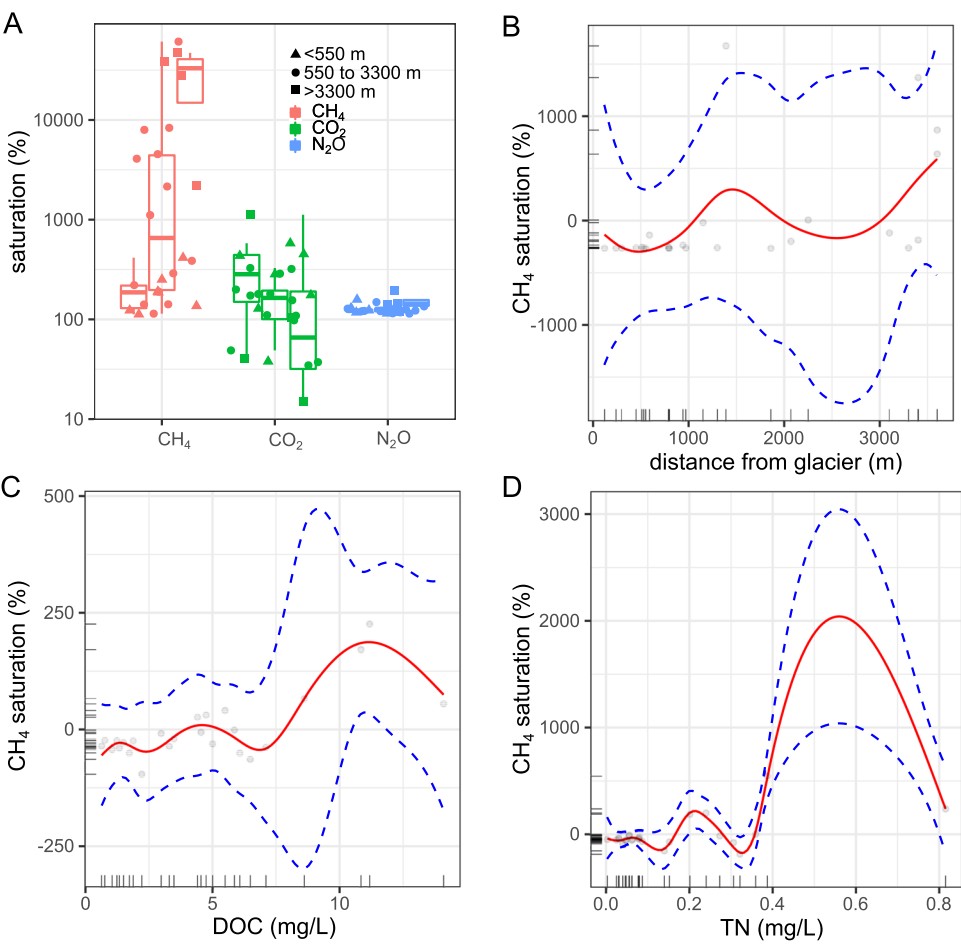

**Fig. 1 | Greenhouse gas (GHG) saturation in the 31 sampled lakes. A** GHGs (log-scale) separated into 3 subsets based on their distance to the nearest glacier. The bottom and top of the box are the 25th and 75th percentile (the lower and upper quartiles, respectively), and the band near the middle of the box represents the median; the upper whisker extends to 1.5 * inter-quartile range (distance between the first and third quartiles). **B, D** Results from generalized additive models (GAMs)

for $CH_4$ saturation along the chronosequences with partial effect splines (as shown by red lines) for significant variables including distance from glacier (**B**), dissolved organic carbon (DOC; **C**) and total nitrogen concentrations (TN, **D**). Standard errors are indicated by dotted lines while data points are shown by open circles. GAMs partial effect splines for significant variables modeling $CO_2$ saturation are shown in Supplementary Fig. 2.

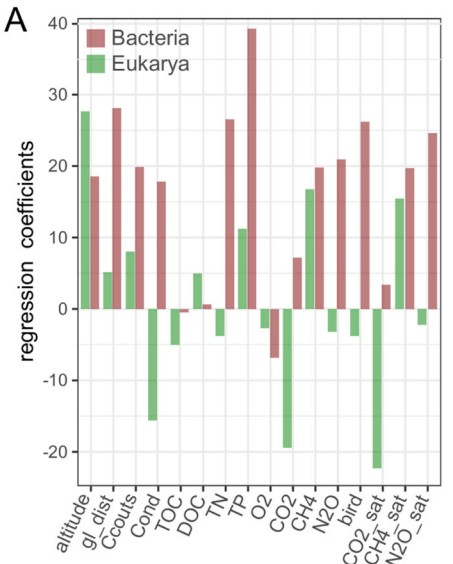

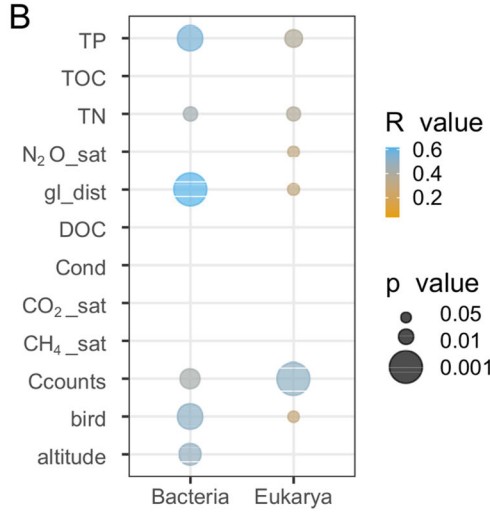

**Fig. 2 | Alpha and beta diversity patterns across the chronosequences.**
**A** Regression coefficients of variables explaining bacterial and eukaryotic richness (as estimated by the Abundance-based Coverage Estimator (ACE) on 16S rRNA gene data) from partial least square (PLS) models. **B** Results of multiple correlations using environmental variables and beta-diversity ordinations (non-parametric multidimensional scaling - NMDS) of bacterial and eukaryotic 16S rRNA gene data to evaluate community organization across measured gradients. Only statistically significant ($p < 0.05$) correlations between variables and community composition are given. The size of the bubbles is inverse to the $p$ value while color indicates the correlation coefficient ($R$ value). TP total phosphorus, TOC total organic carbon, TN total nitrogen, O2 oxygen concentrations, N2O N2O concentrations, N2O_sat N2O saturation, gl_dist distance from glacier, DOC dissolved organic carbon, CO2 CO2 concentrations, CO2_sat CO2 saturation, CH4 CH4 concentrations, CH4_sat CH4 saturation, C counts cell counts, bird bird impact, altitude.

the microbial genomes across the five chronosequences using both gene centric (based on annotation and coverage of contigs) and genome centric (based on MAGs) approaches. We assembled (assembly of individual samples) $5.91 \times 10^9$ high quality reads from 29 metagenomic samples, resulting in $5.44 \times 10^7$ contigs longer than 1000 bp with a range per sample from $8.22 \times 10^5$ to $2.98 \times 10^6$ contigs. These contigs captured on average 74% (range 47 to 90%) of the reads per sample, mapping $4.44 \ 10^9$ reads (average $1.52 \times 10^8$ per sample with range from $5.46 \times 10^6$ to $2.47 \times 10^8$ reads), and containing $7.03 \times 10^7$ ORFs. Most domain annotated reads were of bacterial origin (on average 94% of the reads, range 82–99%), while 5.8 (0.87–17.6) % of Eukaryotic and 0.13 (0.003–1.52) % of archaeal origin. Of the ORFs, $1.01 \times 10^7$ and $1.38 \times 10^7$ could be annotated to KEGG and pfam orthology, respectively, encoding major pathways (metabolic traits) such as synthesis of oxygenic and anoxygenic photosynthesis reaction centers, enzymes for carbon fixation, respiration, sulfate reduction, denitrification and nitrification (for more details on data processing results see supplementary table S4–S7). Subsequent co-assembly and binning of contigs resulted in 167 MAGs with 50% completeness and contamination below 5%. Detailed summary statistics including the 167 MAGs are given in supplementary materials (Supplementary Tables 8 and 9).

We observed major changes associated with glacial retreat dissolved N and C concentrations in most key genes for C, N and S cycling as well as for energy metabolism, (Fig. 3). Using gllvm, we reveal a succession towards increased gene content for methane cycling, aerobic respiration and anoxygenic photosynthesis. This coincided with a relative increase in the genetic potential for denitrification with lake succession, mirroring observations of increases in N₂O saturation. Yet, relative genetic potential for nitrification, anammox and N fixation was highest in young lakes, emphasizing the key role of chemolithoautotrophs in these systems (for more detail on the gllvm using trait dynamics see Supplementary Fig. 8).

MAGs retrieved from the lake systems through co-assembly and binning revealed patterns similar to the gene centric approach with heterotrophic and phototrophic genomes often increasing along the lake sequence, while potential chemolithotrophic genomes were

common in lakes highly influenced by the glaciers. This was besides the limited mapping of reads to the 167 MAGs (on average 5.6% with range from 0.2 to 16%) and incompleteness; MAGs used for this analysis had an estimated completeness of at least 70%. Identifying metabolic pathways in individual MAGs, and determining their capacity to contribute to C, N and S cycling (Fig. 4) revealed potential chemolithotrophs *Nitrospirae* (bin668) and sulfate reducing *Verrucomicrobia* (bin45) and Bacteroidetes (bin1514). Typical freshwater bacteria, such as the genera *Polynucleobacter* (bin708), *Methylibium* (bin648), and *Novosphingobium* (bin1427), with synthesis pathways for anoxygenic photosystems, aerobic respiration and carbon fixation pathways[64,65] can be defined as mixotrophs and increased in intermediate succession stages. No potentials for methanogenesis and methanotrophy were identified in MAGs with more than 50% completeness, but in several contigs (see gene centric approach above). Meanwhile, glycoside hydrolases (GHs) that participate in the breakdown of different complex carbohydrates had highest coding densities in MAGs of the *Bacteriodetes* and *Proteobacteria* phyla including orders *Cytophagales, Sphingobacteriales, Chitinophagales* and *Flavobacteriales* as well as *Sphingomonadales* and *Burkholderiales*, respectively. MAGs with high GH coding densities also contained putative degradation pathways for a variety of sugars. These potential polysaccharide degraders were also more abundant in later succession stages with elevated DOC concentrations (Fig. 3A). Preference of these common freshwater lineages for high DOC lakes has been described in previous studies of lake systems[66,67].

## The impact of birds and high organic matter concentrations
The impact of birds is a potentially confounding factor for water quality and microbial communities, since breeding bird populations (notably geese) have increased strongly over vast areas in the Arctic[38,68], but not in the alpine mainland. This increase is primarily accredited to climate change in the high Arctic breeding sites with extended growing and hatching seasons, but also improved overwintering conditions. Geese prefer the more productive, seaward and earliest deglaciated sites for grazing and reproduction, and with extensive grazing on the sparse vegetation and defecation in or near

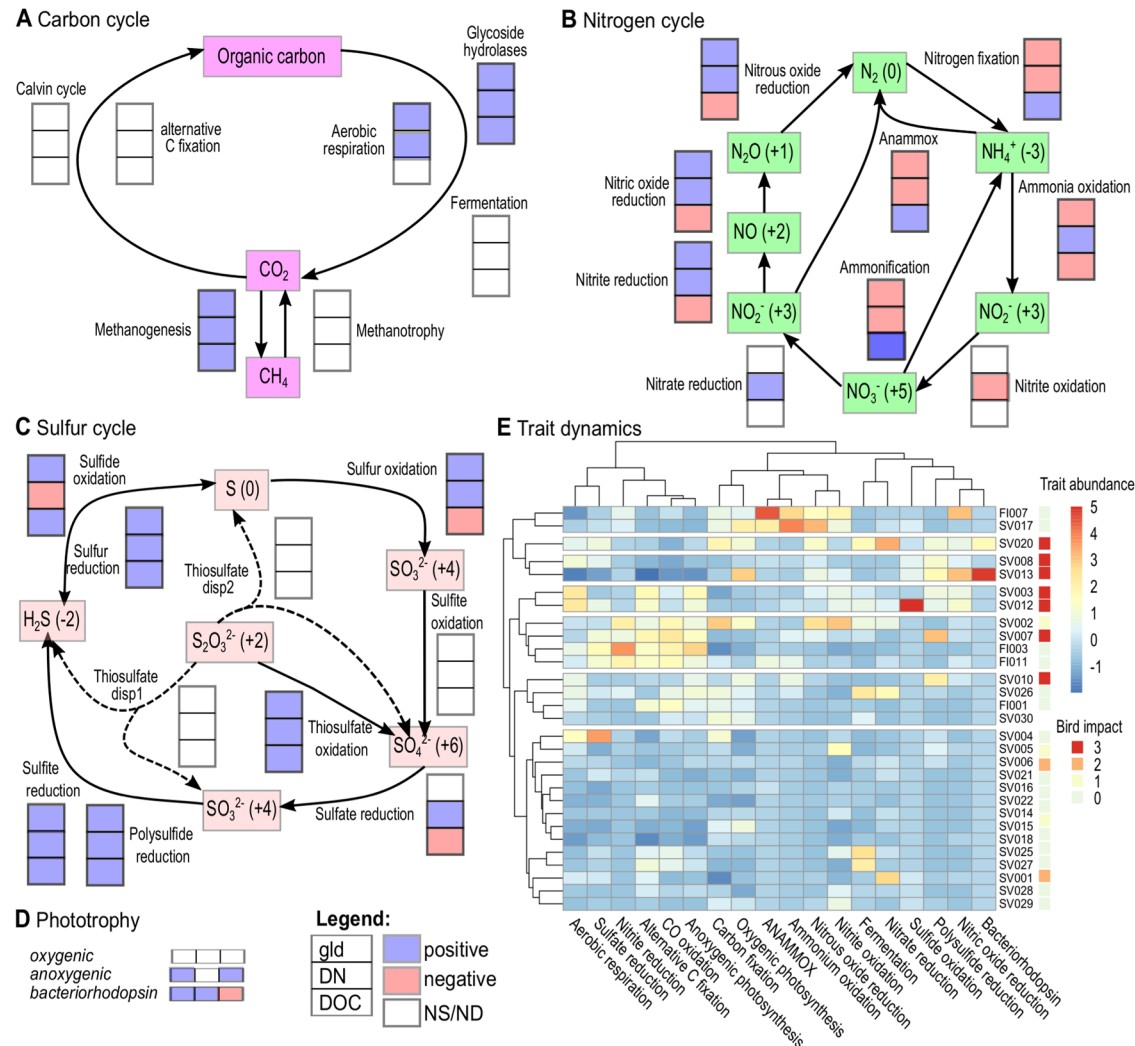

**Fig. 3 | Functional patterns across the five chronosequences as revealed by metagenomics using a gene centric approach.** Fit of generalized linear latent models to responses of individual KEGG gene ontology terms involved in **A** carbon, **B** nitrogen and **C** sulfur cycling as well as **D** phototrophy to glacial distance (gld), dissolved nitrogen (DN) and organic carbon (DOC). Significant relationships are determined from Wald statistics. Filled blue and red boxes represent significant positive and negative relationships with the respective environmental properties. **E** Heatmap of normalized metabolic trait abundances. Vertical dendrogram depicts sample similarities while the horizontal dendrogram shows similarities in distribution of the various traits across the five chronosequences. The latter allows the identification of co-occurring metabolic traits. For example alternative C fixation, CO oxidation and anoxygenic photosynthesis show highly corresponding abundance patterns and cluster tightly. Bird impact (0–3 indicating none, low, medium and high presence of birds and/or droppings) by sample is depicted by the color coding on the far right. Detailed results from the generalized linear latent models are given in Supplementary Figs S7 and S8.

adjacent ponds and lakes, they fertilize these systems substantially[38]. Functional resemblance in bird impacted sites was reflected by the similarity of a newly formed thermokarst thaw pond (SV013) with two freshwater lagunes (SV008 and SV020) (Fig. 3E; see also Supplementary Fig. 8 for KEGG dynamics). In these three systems a high contribution of bacteriorhodpsin and N cycling genes (i.e. genes involved in anammox and nitric oxide reduction) was encoded in the metagenomes, with the shoreline lagunes having additionally a high share of their genomic content encoding for S cycling, most likely reflecting the marine influence. MAGs with high read recruitment in the newly formed pond encoded mostly for aerobic respiration with some MAGs including sets of genes for fermentation, nitrate reduction and nitrite oxidation (Fig. 4).

Three other highly bird impacted systems (SV003, SV007 and SV012) with relatively high concentrations of dissolved organic carbon had divergent functional profiles with a high degree of genes indicative for N cycling, aerobic respiration and anoxygenic photosynthesis and non-Calvin cycle based carbon fixation. In these systems high read

recruiting MAGs encoded for aerobic respiration and anoxygenic photosynthesis (bin324 and bin921). These diverse functional profiles suggest that bird impact and increasing nitrogen availability[38] are associated with different functional succession paths discriminated by oxygenic and anoxygenic photosynthesis. The positive association in the contribution of anoxygenic phototrophs with DOC concentrations reflects observations from boreal forest systems at lower latitudes and altitudes[69]. It suggests a trend toward photoheterotrophic bacteria[70,71] with increasing allochthonous organic matter inputs with optical properties selecting for high (far red) wavelengths with low energy yield per proton, though still capable of modulating carbon cycling in freshwater systems[72].

## Short retention times in proglacial systems reflect supraglacial, englacial and subglacial communities

Systems close to the glacier (proglacial systems, closer than 550 m) with short retention times are fed by water that percolates and flows through a network of hydraulically linked fractures[73] and cavities from

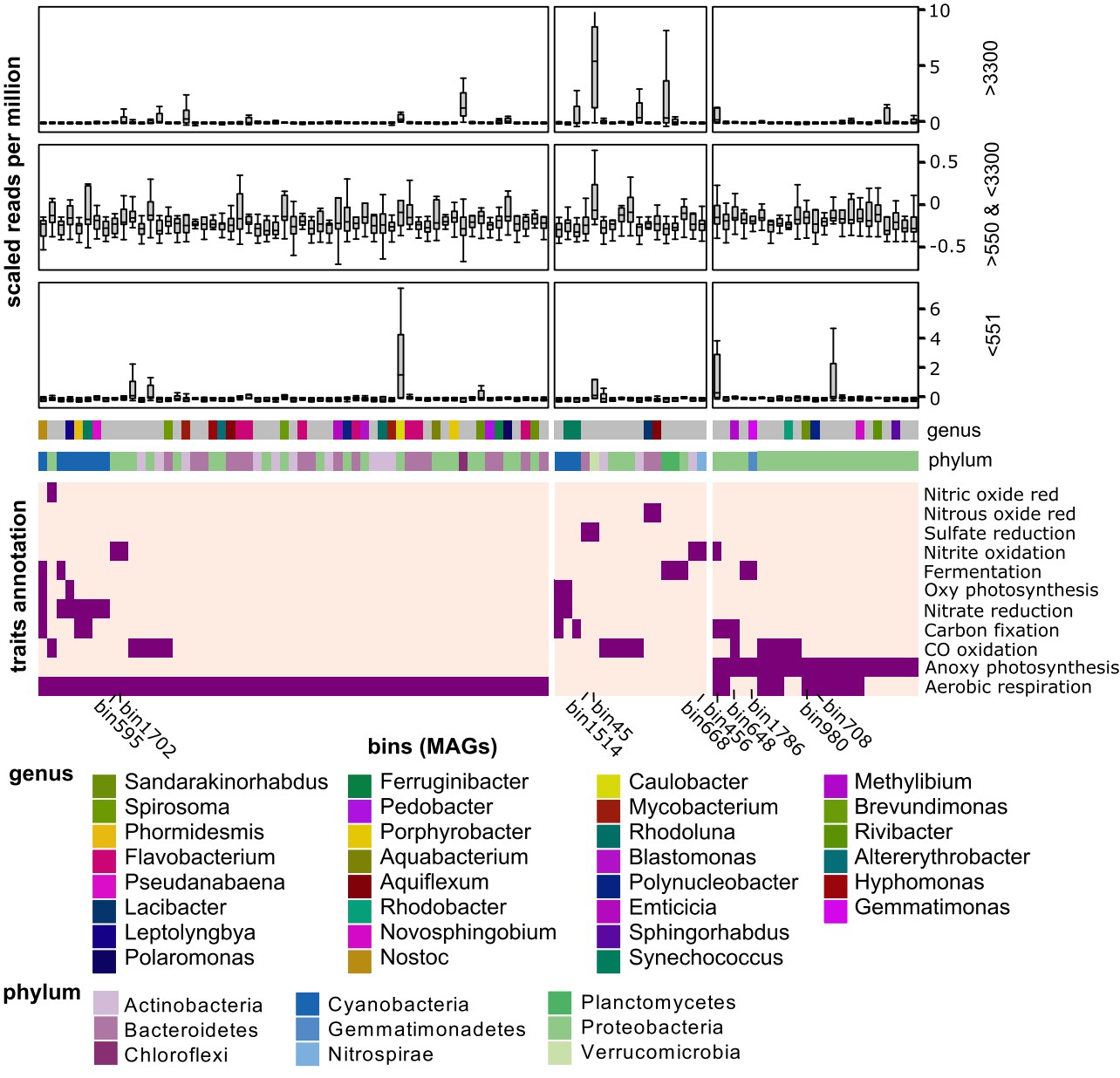

**Fig. 4 | Metabolic potential and dynamics of metagenomes assembled genomes (MAGs) across the chronosequences.** The taxonomic affiliation of the MAGs at the phylum and genus level, as well as MAGs mentioned in the text are given. In the third panel the read coverage of the MAGs across the chronosequences is depicted by separating the system into three categories depending on their distance to the closest glacier. The bottom and top of the box are the 25th and 75th percentile (the lower and upper quartiles, respectively), and the band near the middle of the box represents the median; the upper whisker extends to 1.5 * inter-quartile range (distance between the first and third quartiles).

the supraglacial surface, through the englacial zone and subglacial systems[74]. These proglacial lakes are thus characterized by low nutrient and organic carbon concentrations, as well as chemolithotrophic and phototrophic microbes discharged from the glacier. We identified a diverse set of functional traits characterized by a high contribution of genes encoding for nitrogen cycling (Fig. 3E). In particular genes indicative for metabolic traits such as aerobic and anaerobic ammonium oxidation, and N fixation were associated with proglacial lakes (Fig. 3B). N fixation has been quantified in glacial ponds (i.e. cryoconite holes) although providing only a small fraction of the total N inputs[75]. Potentially nitrifying MAGs were identified in these systems including autotrophic *Nitrospirae* (bin668) and potential heterotrophic nitrifiers including beta Proteobacteria (bin595 and bin1702). This was despite dissolved and total nitrogen concentrations being very low, not

exceeding 0.031 mg L$^{-1}$. The low N$_2$O concentrations in these systems further suggest that nitrogen cycling is low in these proglacial systems and that observed gene inventories including dissimilatory sulfate reduction and carbon fixation, that point towards chemolithotrophic communities, are likely discharged from subglacial sources.

The association with lakes at the glacier forefront fits with the hypothesis that chemolithoautotrophic processes are important in sub-glacier associated systems[35]. Contrary, signals for oxygenic photoautotrophs, as provided by 18S rRNA gene sequences, are likely originating from supraglacial communities such as snow, cryoconite holes and ice surfaces. It can be speculated that allochthonous microbes originating from the catchment decrease with increasing age of the lake systems as productive and retention times increase across the chronosequences.

## Functional compositions in GHG supersaturated systems

High methane concentrations exceeding $0.9\,\mu M$ in the surface, and thus representing conditions of high $CH_4$ emissions (oversaturation in the 10th of thousand %)[76], were observed in SV007, SV008, SV012, and SV026. Similarly, the newly formed permafrost pond (SV013) had concentrations of $CH_4$ at 82 nM (estimated oversaturation of 2211%) and exhibited highest measured concentrations of $CO_2$ (3.1 mM) with an estimated oversaturation of 1121%. Highest supersaturation of $CH_4$ was observed at site SV026 (60,000%), similar to high values of supersaturation observed previously in similar systems[51], and supported by observations of constant bubble release that likely originates from coal deposits beneath the lake. Geologic methane seeping through the edges of thawing permafrost and receding glaciers originating from coal beds has been observed previously[13,77,78] and can contribute substantially to local and global $CH_4$ emission[79,80].

Despite the truly extreme methane concentrations in the surface[76], the lakes possessed low proportions of potential methanotrophs (mean proportion 1.1% with range from 0 to 7.2%), such as *Methylobacter, Ca. Methyloumidiphilus, Methylovulum* and *Methylosoma* (Supplementary Fig. 9). Methane oxidation rates measured in some of these systems (SV007, SV008, SV012) ranged from 0.002 to $0.385\ nM\ h^{-1}$ which is at the low end of rates observed in lake systems[81]. The shallowness and rapid mixing of the wind-exposed arctic and alpine lakes leads to short residence times of $CH_4$ which does not favor biogenic methane oxidation in the water column. Together with the paucity of metagenomic functional annotations (coverage information), and the low methane oxidation rates, this suggests that the water column of arctic lakes does not oxidize large amounts of $CH_4$ during the arctic and alpine summer. Low methane oxidation potentials have also been found in small boreal lakes[82] where most of the $CH_4$ produced in anoxic parts of the water column and sediments is emitted to the atmosphere[83].

## Considerations of functional and genomic diversity across the landscape

To gain deep insights into the regulation of ecosystem functions, understanding co-occurrences of key functions, genes and species in ecosystems and genomes is an important prerequisite[84,85]. System-wide data such as provided in our study enable a systematic and knowledge guided view on different microbial community functions and/or structures. A first insight can be provided by co-occurrence analysis of metabolic traits across the deglaciated landscapes. Using hierarchical cluster analysis, we unraveled associations among traits such as between anoxygenic photosynthesis and non-CBB based carbon fixation (e.g. the reductive citric acid cycle and 3-hydroxypropionate cycle) (Fig. 3E). This is expected from analyses of bacterial genomes and is further confirmed by our genome centric analysis of MAGs (Fig. 4). This includes potential mixotrophy as suggested by genome reconstructions of the genera *Polynucleobacter* (bin708), *Sandarakinorhabdus* (bin980), *Novosphingobium* (bin1786) and others (Fig. 4). Coupling of nitrification, anaerobic ammonium oxidation (anammox), nitrate reduction and dissimilatory sulfate reduction, as suggested by observed co-occurrences in the communities (Fig. 3E) but also by previous studies[67,86], emphasizes a coupling of N and S cycling through chemolithotrophic processes.

The metagenomic data unraveled general dynamics of dominant metabolic pathways throughout the successional stages and the community response to deglaciation. As shown in Fig. 4, there is a high variability in the distribution of genomes across the landscape. Read coverage, despite having a limited mean read coverage of just 5.6% (range 0.2 to 16%) by the 167 MAGs across all samples, indicated a unique genome composition in each system. This is further corroborated by looking at coverage information of obtained contigs across the chronosequences where most contigs were highly abundant in only a single system. By contrast, the gene centric approach revealed functional succession with clear trends in C, N and S traits (Fig. 3). These trait dynamics were associated with taxonomic composition including both 16S (Spearman correlation R = 0.52, $p < 0.001$) and 18S rRNA (Spearman correlation $R = 0.53$, $p < 0.001$) amplicon data. Such association between taxonomic and functional diversity has been controversial in water biomes, largely because of the probable prevalence of functional redundancy[87]. However, previous studies on this topic used a relatively coarse resolution of ecosystem functioning[30,88,89], potentially inflating the estimated functional redundancy, while we used highly-resolved functional traits (i.e. KEGG annotations) and taxonomic (i.e. amplicon sequence variants) diversity data.

Close inspection of the functional diversity as visualized by NMDS plots (Supplementary Fig. 3C) and hierarchic clustering based on landscape resolved metagenomics (Fig. 3E) indicate homogenous succession patterns in the various chronosequences, in particular when comparing the chronosequences obtained from Finse and Svalbard. This suggests that environmental drivers of functional diversity are similar across deglaciated landscapes and contrasts ASV and taxonomy-based succession patterns where samples clustered according to their geographic origin (Supplementary Figs. 3A, B). More consistent patterns in functional than taxonomic diversity across vastly scattered sites is a common feature in nature and has been linked to functional redundancy among taxa[90,91] and dispersal limitations[92,93]. As such, functional diversity has also been proposed to have better predictive power for ecosystem functions[94,95] such as GHG emissions.

To conclude, our chronosequences offer a window into the taxonomic and functional succession of microorganisms in vastly changing alpine and arctic landscapes and provide genomic evidence for the capabilities of microbes in proglacial freshwater biogeochemical cycling, including greenhouse gas emissions. We reveal a shift from chemolithotrophy to phototrophy encoded in microbial genomes with allochthonous organic carbon inputs likely regulating the balance between oxygenic and anoxygenic photosynthesis as well as the contribution of heterotrophic bacteria. Nitrogen inputs by birds were clearly reflected in the functional metagenomic profiles and our microbe-centric analyses identified abundant microbial taxa in proglacial lakes that likely play important roles in nitrogen and carbon transformations and the release of GHGs from the water column.

## Methods

### Sampling and sampling sites

We studied five glacial chronosequences at Svalbard, Kongsfjorden (79°N) and Longyearbyen (78°N), and mainland Norway (Finse, 60°N). Our design is based on the assumption that space can substitute time, which implies that every site has essentially the same biotic and abiotic history[42,96]. This is not a new approach; chronosequences are commonly used to study succession[42,97–99] and they are particularly useful if the changes of interest occur over decades or centuries[21,42]. Chronosequences are also useful when stochastic events and disturbances at the sampling locations are a regularity[42,100]. High Arctic glacier forelands are comparatively unstable due to the thermal and hydrological structure of the polythermal glaciers commonly found in the High Arctic[101,102], and can thus be used as chronosequences in the sampling design.

We selected lakes ($n = 31$) with minimal anthropogenic influence, spanning a distance of 0.2 to 10 km from the glaciers. Samples were taken during summer 2019 (8th to 11th of August at Svalbard and 23rd of August at Finse). Some study sites (SV001-003, and SV007) were close to human settlements (Ny-Ålesund and Longyearbyen) and highly impacted by birds. Chronosequences differ in terms of the extent of glaciation, aspect, elevation, river systems, presence of lakes and types of lakes. Lakes in the upper parts (< 550 m from glacier) are characterized by glacier meltwater and short retention time with no

vegetation in the surrounding. In the lower parts of the catchment (550–3300 m), the bedrock is covered with unconsolidated Holocene materials: moraines, glaci-fluvial and marine deposits with some form of vegetation. The lowest parts are flat and characterized by mixed glaci-fluvial material and permafrost covered with tundra vegetation. Retention times are longest in these lake systems. In three of the chronosequences, lagunes were included which have been formed by moraines and marine material.

### Chemical analysis

Water temperature was measured on site. Water samples were collected from at least three locations with a 4 meter long grabber and pooled prior to subsampling for DNA, nutrients, bacterial numbers, and chromophoric characteristics of the water. For gas analysis, 30 ml of water were carefully taken from the lake directly with a 60 ml Luer Lock syringe equipped with three-way stopcocks. Subsequently, a 20 ml ambient air headspace was created in each syringe and the content was acidified with 0.6 ml 3% HCl (final concentration 0.06% HCl)[82]. Syringes were then closed and equilibrium was reached in the headspace at field temperature by shaking them for 3 min stopping for 30 s each at 1 min intervals. Thereafter, 15 ml headspace gas were transferred to 12 ml evacuated serum vials crimp sealed with 10 mm butyl septa. For gas chromatographic analysis, 2 mL of headspace gas were sampled (autosampler GC-Pal, CTC, Switzerland) and injected into a GC with He back-flushing (Agilent 7890 A, Santa Clara, CA, USA). The GC was equipped with a 20 m wide-bore (0.53 mm) Poraplot Q column operated at 38 °C with He as carrier gas for separation of GHGs from bulk gases (i.e. Ar, $O_2$ and $N_2$). A 60 m molsieve column was used to separate Ar, $O_2$ and $N_2$. $N_2O$ was measured with an electron capture detector (ECD) operating at 375 °C using Ar/$CH_4$ (80/20) as make-up gas, and $CH_4$ with a flame ionization detector (FID). $CO_2$, $N_2$ and $O_2$ were measured with a thermal conductivity detector (TCD). Certified standards of $CO_2$, $N_2O$, and $CH_4$ in He were used for calibration (AGA, Germany), whereas $N_2$ and $O_2$ were calibrated against air. The analytical precision for all gases was better than 1%. Saturations relative to atmospheric equilibrium were calculated for all gases according to in situ water temperature measurements. Henry's law constants, taken from[103], were recalculated from 20 °C to water temperature using the Clausius–Clapeyron equation. Then, equilibrium concentrations at in situ temperatures were calculated based on Henry's law constants and average atmospheric partial pressures of the individual gases.

### Water incubations

Briefly, three incubation replicates were prepared for a selected number of sites (SV007, SV008, SV012) in the field. For each replicate, 80 ml of water were transferred into a 120 mL serum bottle with a 60 ml syringe equipped with a tube to avoid potential disturbances (e.g. outgassing), before sealing them with a rubber septum keeping ambient air as headspace. The bottles were covered with aluminum foil to protect them from light, transported to the laboratory and incubated for 24 h at 4 °C. The incubations were stopped by adding 1.4 ml 3% HCl (final concentration 0.06%) and then shaken for 3 min to reach gas equilibrium in the headspace. Gas concentrations were obtained as described above. $CH_4$ consumption per hour was estimated by subtracting the concentrations reached at the end of the incubations from measured in situ concentration and dividing the results from the subtraction by 24 h.

### Nutrient concentrations

Total and dissolved phosphorus and nitrogen concentration were measured using standard methods. Total and filtered P was measured on an auto-analyzer as phosphate after wet oxidation with peroxodisulfate. TN was measured as nitrogen monoxide by chemiluminescence using a TNM-1 unit attached to the Shimadzu TOC-VWP analyzer (Shimadzu Corporation, Japan). Total organic carbon (TOC)

concentrations, also known as non-purgeable organic carbon (NPOC), and dissolved organic carbon (DOC) concentrations on 0.2 μm filtered samples were obtained by analysis on a Shimadzu TOC-L with sample changer ASI-L (Shimadzu Corporation, Japan).

### Bird impact

As an indicator of the fecal input from birds into the ponds the number of bird droppings was counted at a distance up to approximately 25 m from the shoreline. The density of droppings was recorded using a semi-quantitative scale (none, low, medium, and high) and converted to an ordinal scale ranging from 0 to 3 for statistical analysis[36]. However, such estimates only provide insight into the number of visiting birds at a relative short time scale and, in addition, a significant proportion of goose droppings can be consumed by reindeer, which may strongly bias the estimates of the density of droppings[36]. Since there was a significant relationship with the density of droppings with relative count estimates of *Clostridiaceae* (a family representing a potential bird fecal indicator; others such as *Enterococcaceae* and *Fusobacteriaceae* had too low prevalence) (Spearman correlations R = 0.60, $p < 0.001$) from amplicon sequencing data, as well as dissolved (Spearman correlations $R = 0.53$, $p = 0.002$) and total (Spearman correlations R = 0.60, $p < 0.001$) nitrogen concentrations, we have confidence in the droppings based bird impact proxy. A potential cofounding factor could be that most highly bird impacted sites were close to human settlements.

### Bacterial (prokaryotic) numbers

Samples for bacterial counts were fixed with 37% borax buffered formaldehyde (final concentration 2%) and stored at 4 °C prior to analyses. Cells were stained with the fluorescent nucleic acid stain SYBR green I (Molecular probes, Invitrogen, Waltham, Massachusetts, USA) for at least 30 min (1x final concentration) and counted with an Attune® NxT Acoustic Focusing Cytometer (Thermo Fisher) equipped with an Invitrogen Attune NxT Autosampler and a 488 nm laser using green fluorescence for triggered particle scoring. Cell counts were extracted using Attune Nxt Software (Thermo Fisher) and using a gating strategy as outlined in Supplementary Fig. 10.

### DNA sampling and extraction

For DNA, between 0.2 and 1.2 liter of water was filtered through 0.2 μm Sterivex cartridges (Merck Millipore, Germany), in duplicate, by using sterile syringes. Samples were immediately frozen on-site in dry shippers and transported to the laboratory. Upon arrival samples were stored in −80 freezers until further analyses. Total DNA was extracted using a DNesay PowerWater Sterivex Kit (Qiagen, Germany).

### Amplicon sequencing of bacteria and eukaryotes

Bacterial and eukaryotic SSU rRNA gene amplicons were sequenced on a MiSeq machine (Illumina, San Diego, California, USA) following procedures from[57]. In short, we used bacterial primers 515FB 5'-GTGY-CAGCMGCCGCGGTAA-3' and 806RB 5'-GGACTACNVGGGTWTCTAAT-3' and eukaryotic primers TAReuk454FWD1 5'-CCAGCASCYGCGG-TAATTCC-3' and V4 18S Next. Rev 5'-ACTTTCGTTCTTGATYRATGA-3'[104] equipped with Nextera adapters and indices. The pooled samples were sequenced at IMR sequencing facility (Dalhousie University, Halifax, Canada) using an Illumina MiSeq with a 2 × 300 bp chemistry.

### Shotgun-metagenomic sequencing

Libraries were prepared from 20 ng input DNA using the Nextera DNA Flex reagents (Illumina, San Diego, California, USA) according to manufacturer's instructions, with 8 cycles PCR and unique dual index adapters. Library concentrations were determined by quantitative PCR using KAPA reagents (Roche) and a single equimolar pool of all libraries prepared. Samples FI016 and SV011 did not give any libraries with acceptable quality, and thus these samples were excluded from

metagenome sequencing. Sequencing was performed on an Illumina NovaSeq S4 flowcell with 150 bp paired end reads.

## Bioinformatic analyses

**Amplicon sequence processing.** After raw sequence data had been trimmed of primers with CUTADAPT[105] and sequences without matching primers removed, they were analysed with R package dada2[106] for de-replicating, denoising and sequence- pair assembly. After manual inspection of quality score plots, the forward and reverse reads of the bacterial amplicons were trimmed at 250 and 210 bp length respectively and the eukaryotic forward and reverse reads at 270 and 190 bp length, respectively. Additional quality filtering removed any read pairs with a single phred score below 20. After reads had been dereplicated, forward and reverse error models were created in dada2 with a subset of at least $10^7$ bases. For respective ASV tables, chimeras were removed using 'removeBiomeraDenova' in the dada2 package, and only bacterial and eukaryotic sequences were kept after assigning taxonomy using the Naïve Bayesian classifier in combination with the SILVA138 (for bacteria) or PR2 (for eukaryotes) databases.

**Metagenomic analysis.** For metagenomic analysis we used the fully automated pipeline SqueezeMeta[107] (Version 1.5.1, Jan 2021) including quality filtering using Trimmomatic[108], individual sample read assembly and co-assembly with Megahit[109] and mapping with Bowtie2[110]. Annotations were performed against KEGG[111] and Pfam[112] databases using diamond[113]. Co-assemblies were used to construct MAGs using various binners and DAS tool[114].

**Metabolic reconstructions.** Metabolic functions of the microbial communities were reconstructed using two different approaches. Community metabolism was assessed using assembled contig coverage (transformed into TPMs) and gene annotations. KO annotation was used for functional analysis and was focused on the two main biogeochemical cycles for this type of lakes, that is, carbon (C), and nitrogen (N) cycling. The genetic potential of the microbial community was analyzed following the C, N, and S marker genes (KOs) as reported by[115,116] with a few modifications, in particular adding markers for photosynthetic microorganisms and hydrolases (for details see R code in https://github.com/alper1976/chronosequences). We also used the pathways encoded in high-quality MAGs and coverage information to infer metabolic variations along the chronosequences.

**Ecoinformatic analysis.** Rarefaction curves were computed using 'rarecurve' from the vegan package with outputs suggesting that ~9,000 bacterial and ~20,000 eukaryotic reads are sufficiently covering ASV richness. Rarefying removed 1217 bacterial and 222 eukaryotic ASVs as well as two samples (SV011 and SV021), and resulted in 9229 bacterial and 21,348 eukaryotic reads per sample. For shotgun metagenomic data we used TPMs which are based on proportions calculated over an arbitrary count total (in this case, the number of reads sequenced per sample) and as such are compositional (Gloor et al. 2017). Alpha diversity indices were estimated from rarefied ASV tables including species richness with ACE, a nonparametric method for estimating the number of species using sample coverage and Pielou´s evenness. We used maximal information-based nonparametric exploration (MINE)[58] to capture relationships from spatial patterns of planktonic taxa using relative abundance matrices. This nonparametric approach also identified nonlinear relationships among pairs of taxa (bacteria and eukaryotes), providing some clues to the interactions among microbes.

**Statistical analysis.** Generalized linear models (GLMs) and generalized additive models (GAMs) of the gaussian family to the dependent variables were run in R using the 'mgcv' package[117]. To test the dependency of $CO_2$, $CH_4$ and $N_2O$ saturation ratio, we used a gam model with "REML" as the smoothing parameter method on each of the explanatory variables DOC (mg $L^{-1}$), TP (µg $L^{-1}$), TN (mg $L^{-1}$), glacial distance (gl_dist; m) and conductivity (EC; µS $cm^{-1}$). Predictive variable selection was done by applying additional shrinkage on the null space of the penalty with the select=TRUE argument in the mgcv::gam function, as recommended by[118]. Partial least square regression analysis (PLS) was used to analyze relationships between scaled alpha diversity estimates and environmental variables using cross-validation. For ordinations (i.e. NMDS and RDAs/CCAs) vegan was used with Hellinger standardization and Bray Curtis as the distance measure on the compositional matrices (beta-diversity measures) including taxonomy (i.e. class, genera), gene ontologies (KEGGs) and ASVs. Environmental vectors and factors were then fitted onto the ordination to obtain correlation co-efficient and significance levels with beta-diversity.

The gllvm function of the gllvm package[119] was used to fit generalized linear latent models of the negative binomial type to responses of individual ASV, KEGG gene ontology terms and pathways to the explanatory variables DOC (mg $L^{-1}$), TP (µg $L^{-1}$), TN (mg $L^{-1}$), glacial distance (gl_dist; m) and bird impact, with relationship significance determined from Wald statistics.

## Reporting summary
Further information on research design is available in the Nature Portfolio Reporting Summary linked to this article.

## Data availability
The raw demultiplexed sequence data (amplicon and metagenomic reads) generated in this study have been deposited in the Sequence Read Archive (SRA) under BioProject accession code PRJNA729725. The ASV tables, metadata and method descriptions have been deposited in the OSF data repository (www.osf.io) under https://doi.org/10.17605/OSF.IO/PNWKS.

## Code availability
The code used in this study is available under https://github.com/alper1976/chronosequences with release v1.0.0 published under DOI: 10.581/zenodo.7886432.

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

## Acknowledgements

Funding was provided through project ARCTIC-BIODIVER financed by Belmont Forum-BiodivERsA and The Research Council of Norway to D.O.H., and Arctic Field Grant 310631 to N.V., also funded by The Research Council of Norway. We want to thank Berit Kaasa for performing the nutrient analyses. Access to Attune Nxt flow cytometer was provided by the Flow Cytometry Core Facility at Oslo University Hospital, Gaustad. The sequencing service for metagenomics was provided by the Norwegian Sequencing Centre (www.sequencing.uio.no), a national technology platform hosted by the University of Oslo and Oslo University Hospital and supported by the "Functional Genomics" and "Infrastructure" programs of the Research Council of Norway and the Southeastern Regional Health Authorities. Computational analysis was performed on UNINETT resource SAGA (https://documentation.sigma2.no/hpc_machines/saga.html) under projects NN9744K and NN9745K, and for long term data storage we use the National Infrastructure for Research Data (NIRD) under project NS9745K.

## Author contributions

A.E. and D.O.H. developed the study. L.F., D.O.H. and A.E. collected survey samples. P.D. conducted geochemical analyses. A.E. processed metagenomes, recovered MAGs, and performed statistical analysis. A.E. reconstructed metabolic potential of MAGs. J.W. prepared amplicon libraries and processed amplicon data with A.E. performing most of the statistical analyses. N.V. designed and conducted rate measurements with support from L.F. and P.D. J.W. and A.E. performed the flowcytometry analysis. A.E. devised the figures (with the exception of figures S1 and S3 which were devised by J.W.) and wrote the initial manuscript with input from J.W. and D.O.H. The final version was edited and approved by all authors.

## Competing interests

The authors declare no competing interests. AE is the founder and co-owner of eDNA solutions AB, Sweden.
