## [Peer Review File · Nature Communications]

Trajectories of freshwater microbial genomics and greenhouse gas saturation upon glacial retreatReviewer #1 (Remarks to the Author):

The authors substitute space for time and examine microbial communities in several chronosequences of lakes. The authors observe patterns indicative of succession and shifting communities due to the presence of birds. I have two main concerns about the paper:

The authors invoke dispersal from glacier surfaces but still present the findings as if these communities are endemic to the lakes. These points should be clarified.

The authors offer too much speculation regarding the The metagenomic data, particularly the genomes (MAGs), when so few were recovered.

Line 32: The methods suggest lakes impacted by birds was not common in the data set. Can you reconcile this statement with the methods (Line 397 - Lakes SV001-003, and SV007?).

Line 82: expected [to] drive

Line 216: Why not bin the individual metagenomes?

Lines 280-299, Lines 329-334: If proglacial lakes are receiving a large input of glacier microbes, how can you discern between endemic microbes vs. transported cells etc from the glacier? This seems like an key distinction to be able to make to compare the proglacial lakes (and their endemic microbial communities) to other lakes along the chronosequence.

Line 302,314: Please provide references.

Lines 327-358: This section seems highly speculative and so few details are provided that link the speculation back to actual data and observations provided in the paper. The reader has the impossible task of trying to link geochemistry and MAGs to individual lakes across the chronosequences. Regardless, this section is far too speculative as presented and is not bolstered by the (DNA) data provided. Furthermore, what evidence is that that the metagenomic approach provided sufficient sequence depth to conclude that there is high variability in genomes? Especially given that the co-assembly resulted in only one high quality bin (>90% complete). Very little (to no) evidence is provided that there is high genomic diversity across the landscape.

Methods

Please provide a table of geochemistry and gas measurements as well as bird counts.

Please present and discuss the bacterial cell counts.

Please provide more information about the bird transects. How many days (or hours) were the surveys completed on? Why area of lake shore was analyzed? Is being close to human settlements related to the impact of birds? Please clarify.

Amplicon sequencing: Rarefaction resulted in ~15,000 reads per bacterial library (Line 518) but several libraries fall below this threshold (besides SV011 and SV021. For example, SV015, SV017, SV025, SV030. Please double check the analyses to be sure the 15000 reads is accurate.

Figures

Consider a different color palette for Supplementary Figure S3. It is not possible to see any change in distance other than the lightest blue. All other blues look the same shade. Please use a palette with more contrast.

Check Supplementary Figure S4. It is not possible to read the labels.

Reviewer #2 (Remarks to the Author):

In this paper, Wei et al studied 31 lakes at different distances to glaciers in a time-for-space substitution approach. Through several methods, they assess the presence and effect of distance to the glacier, greenhouse gases, and nutrient concentration on the taxonomical composition and metabolic functions of the microbial community of these lakes.

While the subject is interesting and worth studying, I find the paper very confusing with several issues that are problematic and show a lack of rigor in the analyses and the writing.

As an example, at least 2 supplementary figures are missing (S5 and S6), along with supplementary data detailing the analyses. There is also no panel 2C despite the legend.

In the figure 3E, one cannot find which samples are affected by birds, despite this panel being the main support of this part.

Furthermore, several statements and results are based on data that is not shown at all (e.g. glycoside hydrolases presence or the rates of methane consumption) or presented in a way that is very complicated to follow such as the results presented in figure 4. Where there are too many colors representing the different genera and the fact that the names of the MAGs written in the text are not shown in the figure.

Another major point that is not addressed is that while replicates were taken in the field, all were pooled for DNA and chemical analyses which are very problematic especially for chemical values that are sometimes extreme and for amplicon sequencing that is very susceptible to PCR issues.

Finally, the metadata table should probably be placed directly in the supplementary table with a legend detailing the units, indeed, there seems to be confusion with the units of saturation of CO₂ and CH₄ that are sometimes ranging from 1.55 to 291% (I114) and sometimes reach 61000% (I120).

It's quite tricky to follow the "bird" story with the figures shown and thus trust the results presented without extensive changes especially to figure 3E.

Please address the fact that you lack chemical analyses for 6 samples that are further than 5km
The code is not publicly accessible so it cannot be checked.

Other comments

I110, which 7 systems? where is it shown?

Figure 1A, there is probably an issue with the unit as the saturation of CO₂ in the text range from 1.96% to 3.21%, and the data in the figure range from 10 to > 100%

I132-143. the association with specific values of TN and DOC should be better supported by the data.

I176: where is this shown?

I180 - 193: this data is not shown, the figures are missing.

I215: this supplementary material is not available

I216-219: the numbers do not correspond to the ones shown in the supplementary tables

I230, which MAGs, the 16/17 that are of high quality or the 169/200 MAGs with completeness higher than 50%?

I230 - 250, it's impossible to find the specific MAGs in figure 4

I242 - 250, where is the data on glycoside hydrolases shown?

I314, where is this data shown?

Reviewers' comments:

Reviewer #1 (Remarks to the Author):

The authors substitute space for time and examine microbial communities in several chronosequences of lakes. The authors observe patterns indicative of succession and shifting communities due to the presence of birds. I have two main concerns about the paper:

The authors invoke dispersal from glacier surfaces but still present the findings as if these communities are endemic to the lakes. These points should be clarified.

We nowhere talk about endemic communities. We added several sentences to clarify this point.

L177 "system connectivity (downstream sites will receive cells from upstream sites)"

L289- "Systems close to the glacier (proglacial systems, closer than 550 m) with short retention times are fed by water that percolates and flows through a network of hydraulically linked fractures⁷¹ and cavities from the supraglacial surface, through the englacial zone and subglacial systems⁷². These proglacial lakes are thus characterized by low nutrient and organic carbon concentrations, as well as chemolithotrophic and phototrophic microbes discharged from the glacier."

L310- "It can be speculated that allochthonous microbes originating from the catchment decrease with increasing age of the lake systems as productive and retention times increase across the chronosequences."

We also provide a general description of the chronosequences in the methods section to clarify this.

L414- "Chronosequences differ in terms of the extent of glaciation, aspect, elevation, river systems, presence of lakes and types of lakes. Lakes in the upper parts (< 550m from glacier) are characterized by glacier meltwater and short retention time with no vegetation in the surrounding, In the lower parts of the catchment (550m-3300m), the bedrock is covered with unconsolidated Holocene materials: moraines, glaci-fluvial and marine deposits with some form of vegetation. The lowest parts are flat and characterized by mixed glaci-fluvial material and permafrost covered with tundra vegetation. Retention times are longest in these lake systems. In three of the chronosequences, lagunes are included which have been formed by moraines and marine material."

The authors offer too much speculation regarding the The metagenomic data, particularly the genomes (MAGs), when so few were recovered.

We used a definition of 50% completeness and 5% contamination which is in line with previous studies (see for example <https://journals.asm.org/doi/10.1128/mBio.00916-18>; <https://www.nature.com/articles/s41597-021-00910-1>; <https://www.nature.com/articles/s41597-022-01392-5>). Using this cutoff we recovered 167 MAGs recruiting on average 5.6% of the reads per sample, range 0.2 to 16%. This is as expected when comparing to previous studies. We also used a gene centric comparison using recovered contigs larger than 1000 bp to recover a larger functional diversity of the microbes inhabiting the deglaciated landscape. Here, contigs captured on average 74 % (range 47 to 90%) of the reads per sample, mapping 4.44 10⁹ reads (average 1.52 10⁸ per sample with range from 5.46 10⁶ to 2.47 10⁸ reads), and containing 7.03 10⁷ ORFs.

Line 32: The methods suggest lakes impacted by birds was not common in the data set. Can you reconcile this statement with the methods (Line 397 - Lakes SV001-003, and SV007?).

There were many more lakes influenced by birds as those mentioned in the text. These were just examples. We now provide a detailed description of the method used to assess bird impact.

" Bird impact

As an indicator of the fecal input from birds into the ponds the number of bird droppings was counted at a distance up to approximately 25 m from the shoreline. The density of droppings was recorded using a semi-quantitative scale (none, low, medium, and high) and converted to an ordinal scale ranging from 0 to 3 for statistical analysis³⁶. However, such estimates only provide insight into the number of visiting birds at a relative short time scale and, in addition, a significant proportion of goose droppings can be consumed by reindeer, which may strongly bias the estimates of the density of droppings³⁶. Since there was a significant relationship with the density of droppings with relative count estimates of *Clostridiaceae* (a family representing a potential bird fecal indicator; others such as *Enterococcaceae* and *Fusobacteriaceae* had too low prevalence) (Spearman correlations $R = 0.60$, $p < 0.001$) from amplicon sequencing data, as well as dissolved (Spearman correlations $R = 0.53$, $p = 0.002$) and total (Spearman correlations $R = 0.60$, $p < 0.001$) nitrogen concentrations we have confidence in the droppings based bird impact proxy. "

Line 82: expected [to] drive

Fixed

Line 216: Why not bin the individual metagenomes?

We have done this as well but this resulted in very few quality bins, and thus we only present the co-assembly based bins.

Lines 280-299, Lines 329-334: If proglacial lakes are receiving a large input of glacier microbes, how can you

discern between endemic microbes vs. transported cells etc from the glacier? This seems like an key distinction to be able to make to compare the proglacial lakes (and their endemic microbial communities) to other lakes along the chronosequence.

As this was not clear to the reviewer we have added an additional sentence to clarify this.

L310- " It can be speculated that allochthonous microbes originating from the catchment decrease with increasing age of the lake systems as productive and retention times increase across the chronosequences."

We also provide a general description of the chronosequences in the methods section to clarify this.

L414- "Chronosequences differ in terms of the extent of glaciation, aspect, elevation, river systems, presence of lakes and types of lakes. Lakes in the upper parts (< 550m from glacier) are characterized by glacier meltwater and short retention time with no vegetation in the surrounding, In the lower parts of the catchment (550m-3300m), the bedrock is covered with unconsolidated Holocene materials: moraines, glaci-fluvial and marine deposits with some form of vegetation. The lowest parts are flat and characterized by mixed glaci-fluvial material and permafrost covered with tundra vegetation. Retention times are longest in these lake systems. In three of the chronosequences, lagunes are included which have been formed by moraines and marine material."

Line 302,314: Please provide references.

We added Juutinen et al. 2009 which measured CH₄ concentration in a wide range of boreal lakes showing that our CH₄ concentration in the surface represent values in the high range during open water conditions.

Lines 327-358: This section seems highly speculative and so few details are provided that link the speculation back to actual data and observations provided in the paper. The reader has the impossible task of trying to link geochemistry and MAGs to individual lakes across the chronosequences. Regardless, this section is far too speculative as presented and is not bolstered by the (DNA) data provided. Furthermore, what evidence is that that the metagenomic approach provided sufficient sequence depth to conclude that there is high variability in genomes? Especially given that the co-assembly resulted in only one high quality bin (>90% complete). Very little (to no) evidence is provided that there is high genomic diversity across the landscape.

We have removed the high from the heading to put less emphasize on the high variability. Still recruitment of reads to obtained genomes show high variability across the systems indicating a unique genome composition in each system. This is also emphasized by looking at coverage information across obtained contigs and systems (see Supplementary figures S4 and S5; rarefaction curves below indicating good coverages of KEGG dynamics across the sampled lakes). The compositional dynamics should not be mistaken with that of the dynamics of individual genomes as they still might be present across all systems but in much lower abundances; instead, it emphasizes changes in abundance of the most abundant genomes. As such this provides clear evidence for our main conclusion that genomics diversity is highly variable while annotation based functional diversity assessment can reveal consistent community wide patterns.

Figure. Rarefaction curves showing that at around 1,000,000 reads KEGG annotations are close to saturation across the sampled lakes.

Methods

Please provide a table of geochemistry and gas measurements as well as bird counts.

We have added a table with this data to the supplementary material (Supplementary table S10).

Please present and discuss the bacterial cell counts.

We already provide a basic summary on this data and do not see how this can be further expanded.

Please provide more information about the bird transects. How many days (or hours) were the surveys completed on? Why area of lake shore was analyzed? Is being close to human settlements related to the impact of birds? Please clarify.

We now provide a detailed description on how bird impact was assessed.

" Bird impact

As an indicator of the fecal input from birds into the ponds the number of bird droppings was counted at a distance up to approximately 25 m from the shoreline. The density of droppings was recorded using a semi-quantitative scale (none, low, medium, and high) and converted to an ordinal scale ranging from 0 to 3 for statistical analysis³⁶. However, such estimates only provide insight into the number of visiting birds at a relative short time scale and, in addition, a significant proportion of goose droppings can be consumed by reindeer, which may strongly bias the estimates of the density of droppings³⁶. Since there was a significant relationship with the density of droppings with relative count estimates of *Clostridiaceae* (a family representing a potential bird fecal indicator; others such as *Enterococcaceae* and *Fusobacteriaceae* had too low prevalence) (Spearman correlations $R = 0.60$, $p < 0.001$) from amplicon sequencing data, as well as dissolved (Spearman correlations $R = 0.53$, $p = 0.002$) and total (Spearman correlations $R = 0.60$, $p < 0.001$) nitrogen concentrations we have confidence in the droppings based bird impact proxy. A potential co-founding factor could be that most highly bird impacted sites were close to human settlements. "

Amplicon sequencing: Rarefaction resulted in ~15,000 reads per bacterial library (Line 518) but several libraries fall below this threshold (besides SV011 and SV021. For example, SV015, SV017, SV025, SV030. Please double check the analyses to be sure the 15000 reads is accurate.

There was a mix up rarefaction resulted in ~9,000 reads per bacterial library. This is now correctly reported in the methods section.

Figures

Consider a different color palette for Supplementary Figure S3. It is not possible to see any change in distance other than the lightest blue. All other blues look the same shade. Please use a palette with more contrast.

I am color blind and this palette works well for me.

Check Supplementary Figure S4. It is not possible to read the labels.

Labels have been made readable

Reviewer #2 (Remarks to the Author):

In this paper, Wei et al studied 31 lakes at different distances to glaciers in a time-for-space substitution approach. Through several methods, they assess the presence and effect of distance to the glacier, greenhouse gases, and nutrient concentration on the taxonomical composition and metabolic functions of the microbial community of these lakes.

While the subject is interesting and worth studying, I find the paper very confusing with several issues that are problematic and show a lack of rigor in the analyses and the writing.

As an example, at least 2 supplementary figures are missing (S5 and S6), along with supplementary data detailing the analyses. There is also no panel 2C despite the legend.

We have added all supplementary figures and tables to a single file and hope that this will makes it easier for the reviewers to find the information. We have fixed the legend of figure 2 removing the description of panel 2C.

In the figure 3E, one cannot find which samples are affected by birds, despite this panel being the main support of this part.

We have added an additional color scheme to show bird impact.

Furthermore, several statements and results are based on data that is not shown at all (e.g. glycoside hydrolases presence or the rates of methane consumption) or presented in a way that is very complicated to follow such as the results presented in figure 4. Where there are too many colors representing the different genera and the fact that the names of the MAGs written in the text are not shown in the figure.

We have now added ticks and labels to the MAGs mentioned in the text.

Another major point that is not addressed is that while replicates were taken in the field, all were pooled for DNA

and chemical analyses which are very problematic especially for chemical values that are sometimes extreme and for amplicon sequencing that is very susceptible to PCR issues.

In this study we look at gradients, and thus biological replicates are not necessary. Concerning gas data duplicates were measured and variability was minor (less than 5%), hence we used averages from duplicate measurements. Also considering previous results from amplicon studies in our lab it has been shown that technical replicates differ neglectable as well as biological replicates taken from multiple pooled samples of single systems show minor variability (see for example Savio et al. 2015 <https://ami-journals.onlinelibrary.wiley.com/doi/full/10.1111/1462-2920.12886>)

Finally, the metadata table should probably be placed directly in the supplementary table with a legend detailing the units, indeed, there seems to be confusion with the units of saturation of CO₂ and CH₄ that are sometimes ranging from 1.55 to 291% (I114) and sometimes reach 61000% (I120).

The reviewer is correct we forgot to multiple the numbers in the text. The saturation ratios are now given as percentages (multiplied by factor 100) and are not saturation ratios as given earlier. We now also provide the metadata as supplementary table S10.

It's quite tricky to follow the "bird" story with the figures shown and thus trust the results presented without extensive changes especially to figure 3E.

We now provide a detailed description of the bird impact assessment and modified figure 3E.

Please address the fact that you lack chemical analyses for 6 samples that are further than 5km.

The code is not publicly accessible so it cannot be checked.

Now the code is publicly available at <https://github.com/alper1976/chronosequences>.

Other comments

I110, which 7 systems? where is it shown?

These are shown in modified Figure 1A. It's those 7 samples with CO₂ saturation below 100%.

Figure 1A, there is probably an issue with the unit as the saturation of CO₂ in the text range from 1.96% to 3.21%, and the data in the figure range from 10 to > 100%

We have corrected the numbers in the text.

I132-143. the association with specific values of TN and DOC should be better supported by the data.

This is supported by the GAMs as given in Figure 1B-D and Supplementary figure S3 with uncertainties given by standard errors (dotted lines)

I176: where is this shown?

See Supplementary Table S3 showing prevalence of significant co-occurrences. Prevalence (percentage %) of significant co-occurrences between and within kingdoms (Bacteria and Eukaryota) as determined by maximal information-based nonparametric exploration⁵⁴ revealed similarities in inter- and intra-kingdom co-occurrences.

I180 - 193: this data is not shown, the figures are missing.

See supplementary figures S3-S6 showing beta diversity patterns (S3), ASV dynamics (S4) and gllvm models for bacteria (S6) and eukaryotes (S5).

I215: this supplementary material is not available

See supplementary tables S4-S9.

I216-219: the numbers do not correspond to the ones shown in the supplementary tables.

I230, which MAGs, the 16/17 that are of high quality or the 169/200 MAGs with completeness higher than 50%? It is not just completeness but also contamination that resulted in 167 MAGs used for analysis. Numbers should correspond in the new version.

I230 - 250, it's impossible to find the specific MAGs in figure 4

We have now added identifiers for those MAGs mentioned in the text.

I242 - 250, where is the data on glycoside hydrolases shown?

See figure 3A and supplementary figures S8 and S9

I314, where is this data shown?

We now have added a new supplementary figure S9.

Reviewer #1 (Remarks to the Author):

My concerns and suggestions have been addressed.

Reviewer #2 (Remarks to the Author):

While I commend the authors for the modifications that they made to their manuscript which improved its readability, I still have main concerns about this work.

The lack of biological replicates represent a big issue for me. While it's true that gradients sampling could eventually be done without biological replicates, I'd expect much more samples along a transect to consider their presence unnecessary. Indeed, in the paper that the authors cite as their justification, there were 100+ samples along a transect of a single source of water (Danube River) and there were a few biological replicates to confirm the fact that their presence might not be necessary. Here, there is only ~30 samples with 3 different sources of water.

I'm still also extremely confused by the disparity between values of CO₂ and CH₄ saturation given in the text and the values (without any units!!!) in the table S10.

Reviewer #3 (Remarks to the Author):

The authors have an incredible dataset that allows one to characterize the various ongoing changes to glaciers and their respective runoff ecosystems, due to climate change. Using sequence and chemistry information, the change in genomic properties are reported herein. I commend the authors on a well-written and justified draft but do have a few concerns as outlined below that would make it suitable for publication.

Major concerns

- The authors suggest that environmental drivers (L177) may be similar across the deglaciated landscapes across sites. However, this is not reported or highlighted. Maybe they could elaborate further on this point.**
- L154: the authors highlighted the "Microbiome helper" workflow as being used for assessing the doubling of microbial cell numbers and their association with microbial diversity. And suggest a parallel approach was used. However, no details are given for either method. This should be elaborated further both in the results and the methods sections.**
- The representation of the MAGs, their functions are the reads per million is confusing. I would advise the authors to streamline the figure because the RPM values add no value to it. - Importantly, how did the completion of the MAGs affect the true?**
- In a related question, how was the completion of the MAG accounted for during the functional comparison analyses and the gvlms?**
- The representation of the module completion with the heatmap in the gvlm figure would be better off replaced by a figure stating what percentage of MAGs of the total encoded genes for the C, N and S pathways. Did the authors validate their findings for the individual KEGGs with other tools such as METABOLIC and/or Lithogenic which have more curated databases for this type of analysis?**
- L352: Mixotrophy is suggested as a potential mechanism of adaptation within the MAGs, however, the authors do not explain as to how they come to this conclusion. If the evidence is based on the presence or absence of genes, then the completion of the MAGs may cause potential artefacts in this analyses, thus making it a moot point. Please also elaborate on how mixotrophy was arrived upon in this instance.**
- The authors claim to "reveal a shift from chemolithotrophy to phototrophy" in the conclusions. However, the data presented herein is metagenomic, i.e. encoded genes, information. To make such a claim, one would have to demonstrate activity of the said genes which is not provided. I would strongly encourage the authors to tone down this**

language and/or use qRT-PCR on isolated RNA or perform extracellular enzyme activities demonstrating C:N:S levels which may be a better proxy for this shift.

Minor concerns

- 'Chronosequences' should be clearly defined for the non-expert reader, possibly even with a graphic in the supplementary.
- L213: the ORFs generated - were they complete, partial etc.? If either, what was used for the functional annotation?
- The results on the birds' effect on the microbiome seem out of place and need to be nuanced further in the context of the overall findings. Also, the order of the figures in this paragraph is confusing, i.e. going from figure 4 back to figure 3. Maybe it will help the reader to separate out the information to highlight that mentioned in the text. Alternatively, a table may be used.
- Methods - metagenomic analyses: please elaborate on the various parameters and respective version numbers for the various tools and databases used. Indicate also, when the databases were downloaded including version IDs.
- L 554: describe and highlight the "modifications" used for the C, N, S, marker gene analyses
- The code provided should be commented better for reproducibility.

Reviewers' comments:

We want to thank all three reviewers for their valuable comments. This has substantially improved the manuscript.

Reviewer #1 (Remarks to the Author):

My concerns and suggestions have been addressed.

We want to thank reviewer 1 for his valuable inputs on the manuscript.

Reviewer #2 (Remarks to the Author):

While I commend the authors for the modifications that they made to their manuscript which improved its readability, I still have main concerns about this work.

The lack of biological replicates represent a big issue for me. While it's true that gradients sampling could eventually be done without biological replicates, I'd expect much more samples along a transect to consider their presence unnecessary.

An experimental design of replicated sampling of individual systems would cause a number of challenges to the statistical analyses requiring correction of multiple sampling. Though definitely on our mind, obtaining samples from a greater number of systems reached our budget and logistics capabilities. Sampling in the high Arctic is a major challenge for a number of reasons. Most locations are indeed remote and were accessed by foot or bike (multiple day hikes) after travel by air and/or boat, and thus sampling gear and equipment had to be kept at its minimum as this needed to be fitted into three backpacks of the expedition team members (15-30 kg each including a dry shipper); not considering food as well as ammunition and guns to protect the team from polar bears. While the design was aimed as gradient studies along a chronosequence, we will argue that the use of five separate gradient per se also provides a replicate component to the study.

Indeed, in the paper that the authors cite as their justification, there were 100+ samples along a transect of a single source of water (Danube River) and there were a few biological replicates to confirm the fact that their presence might not be necessary. Here, there is only ~30 samples with 3 different sources of water.

Its three main locations but five chronosequences in total – sources were five different glaciers as outlined in the map – supplementary figure S1. This is now clearly stated in the figure legend

"Map of Norway and Svalbard showing the five chronosequences in Ny Ålesund (top insert; 3 chronosequences), Longyearbyn (middle insert; 1 chronosequence) and Finse (bottom insert; 1 chronosequence)."

We like to argue that this is an effective design allowing broad conclusion considering the limitations in resources.

I'm still also extremely confused by the disparity between values of CO₂ and CH₄ saturation given in the text and the values (without any units!!!) in the table S10.

Thanks for pointing out this mismatch. We now provide the units and have adjusted the units to match the figures and tables as well as text throughout the manuscript.

Reviewer #3 (Remarks to the Author):

The authors have an incredible dataset that allows one to characterize the various ongoing changes to glaciers and their respective runoff ecosystems, due to climate change. Using sequence and chemistry information, the change in genomic properties are reported herein. I commend the authors on a well-written and justified draft but do have a few concerns as outlined below that would make it suitable for publication.

Major concerns

- The authors suggest that environmental drivers (L177) may be similar across the deglaciated landscapes across sites. However, this is not reported or highlighted. Maybe they could elaborate further on this point.

That is a good point and we have added modified several sentences to emphasize the generality of the functional succession patterns and its drivers.

"The metagenomic data unraveled general dynamics of dominant metabolic pathways throughout the successional stages and the community response to deglaciation."

"Close inspection of the functional diversity as visualized by NMDS plots (Supplementary figure S3C) and hierarchic clustering based on landscape resolved metagenomics (Figure 3E) indicate homogenous succession patterns in the various chronosequences, in particular when comparing the chronosequences obtained from Finse and Svalbard. This suggests that environmental drivers of functional diversity are similar across deglaciated landscapes and contrasts ASV and taxonomy-based succession patterns where samples clustered according to their geographic origin (Supplementary figure S3A and B). More consistent patterns in functional than taxonomic

diversity across vastly scattered sites is a common feature in nature and has been linked to functional redundancy among taxa^{90,91} and dispersal limitations^{92,93}. As such, functional diversity has also been proposed to have better predictive power for ecosystem functions^{94,95} such as GHG emissions."

- L154: the authors highlighted the "Microbiome helper" workflow as being used for assessing the doubling of microbial cell numbers and their association with microbial diversity. And suggest a parallel approach was used. However, no details are given for either method. This should be elaborated further both in the results and the methods sections.

We do not use the "Microbiome helper" workflow, we determine associations between both alpha and beta diversity and cell numbers (and other environmental variables) using statistical methods such as PLS and NMDS combined with fitting environmental vectors and factors onto the ordination, respectively. PLS was just mentioned in the figure legend previously. We agree with the reviewer that additional information should be provided and modified the methods section accordingly

"Partial least square regression analysis (PLS) was used to analyze relationships between scaled alpha diversity estimates and environmental variables using cross-validation. For ordinations (i.e. NMDS and RDAs/CCAs) vegan was used with Hellinger standardization and Bray Curtis as the distance measure on the compositional matrices (beta-diversity measures) including taxonomy (i.e. class, genera), gene ontologies (KEGGs) and ASVs. Environmental vectors and factors were then fitted onto the ordination to obtain correlation co-efficients and significance levels with beta-diversity."

- The representation of the MAGs, their functions are the reads per million is confusing. I would advise the authors to streamline the figure because the RPM values add no value to it. - Importantly, how did the completion of the MAGs affect the true?

We have modified figure 4 to make it less confusing. NEW FIGURE 4

In addition we have added a disclaimer on the completeness in the results section "This was besides the limited mapping of reads to the 167 MAGs (on average 5.6% with range from 0.2 to 16%) and incompleteness; MAGs used for this analysis had an estimated completeness of at least 70%."

- In a related question, how was the completion of the MAG accounted for during the functional comparison analyses and the gvlms?

The functional comparison is not based on the MAGs – it is based on contigs, and thus represents a gene centric approach. We used both data from the gene centric approach and genomic (based on MAGs) approaches to make inferences on functional changes across the chronosequences. We believe that this is clearly outlined in the manuscript. See Methods "Metabolic reconstructions". This is now also mentioned in the Results/Discussion section of the ms "To obtain spatial resolution functional trait data, we applied a shotgun metagenomics approach and inferred functional changes encoded in the microbial genomes across the five chronosequences using both gene centric (based on assembled contig coverage) and genome centric (based on MAGs) approaches."

- The representation of the module completion with the heatmap in the gvlm figure would be better off replaced by a figure stating what percentage of MAGs of the total encoded genes for the C, N and S pathways. Did the authors validate their findings for the individual KEGGs with other tools such as METABOLIC and/or Lithogenic which have more curated databases for this type of analysis?

This analysis is not based on MAGs. It is based on the gene centric approach. This is now highlighted in the legend of figure 3 to avoid confusion. "Functional patterns across the five chronosequences as revealed by metagenomics using a gene centric approach."

Details on the markers used for trait identification are provided in the R code

https://github.com/alper1976/chronosequences/blob/main/metags/stats_mags.R (L204-) and

https://github.com/alper1976/chronosequences/blob/main/metags/stats_metags.R (L399-)

- L352: Mixotrophy is suggested as a potential mechanism of adaptation within the MAGs, however, the authors do not explain as to how they come to this conclusion. If the evidence is based on the presence or absence of genes, then the completion of the MAGs may cause potential artefacts in this analyses, thus making it a moot point. Please also elaborate on how mixotrophy was arrived upon in this instance.

There are also taxonomic indications as abundant groups of bacteria in these systems are potential mixotrophs – see newly added references – and the gene centric approach on dynamics of anoxygenic photosynthesis genes are a further indication. So this is not purely based on the genomics (MAGs); they provide just a confirmation that these genes or functional potentials can be assembled in genomes.

- The authors claim to "reveal a shift from chemolithotrophy to phototrophy" in the conclusions. However, the data presented herein is metagenomic, i.e. encoded genes, information. To make such a claim, one would have to demonstrate activity of the said genes which is not provided. I would strongly encourage the authors to tone down this language and/or use qRT-PCR on isolated RNA or perform extracellular enzyme activities demonstrating C:N:S levels which may be a better proxy for this shift.

There is clearly a shift towards eukaryotic algae and an increase in oxygen saturation with increasing age supporting the metagenomic data. Still, we modified this sentence to: "We reveal a shift from chemolithotrophy to phototrophy encoded in microbial genomes with allochthonous organic carbon inputs likely regulating the balance between oxygenic and anoxygenic photosynthesis as well as the contribution of heterotrophic bacteria."

Minor concerns

- 'Chronosequences' should be clearly defined for the non-expert reader, possibly even with a graphic in the supplementary.

We have clarified this as well in the introductions " Our study design was based on chronosequences which substitute space for time and thus allow to study ecological succession at decadal time-scales^{21,42}. Samples were obtained along five distinct chronosequences in five individual catchments defined as a series of lakes of different ages formed due to glacial retreat (see map in Supplementary Figure S1)." And modified the figure legend in Supplementary Figure S1 to further outline our study design.

-L213: the ORFs generated - were the complete, partial etc.? If either, what was used for the functional annotation?

ORFs were complete and as outlined in the methods squeezeMeta was used. SqueezeMeta uses diamond to annotate genes – for details see publication on squeezeMeta

<https://www.frontiersin.org/articles/10.3389/fmicb.2018.03349/full>. We provide this detail in the methods section and refer to the most recent publication presenting diamond.

"For metagenomic analysis we used the fully automated pipeline SqueezeMeta¹⁰⁷ (Version 1.5.1, Jan 2021) including quality filtering using Trimmomatic¹⁰⁸, individual sample read assembly and co-assembly with Megahit¹⁰⁹ and mapping with Bowtie2¹¹⁰. Annotations were performed against KEGG¹¹¹ and Pfam¹¹² databases using diamond¹¹³. Co-assemblies were used to construct MAGs using various binners and DAS tool¹¹⁴."

- The results on the birds' effect on the microbiome seems out of place and needs to be nuanced further in the context of the overall findings. Also, the order of the figures in this paragraph is confusing, i.e. going from figure 4 back to figure 3. Maybe it will help the reader to separate out the information to highlight that mentioned in the text. Alternatively, a table may be used.

Admittedly, the bird effect in a climate change context was not well explained, and hence this is now implemented in the revision (L 278-284), yet it was already mentioned in the introduction (L 92). In brief the phenomenon of expanding and increasing bird (notably geese) population has taken place in many parts of the Arctic, primary related to climate change with improved breeding conditions (increased growing and hatching season, but also improved overwintering conditions. The impact of the local environment by grazing and defecation (fertilization) is substantial and given that this is another, potentially confounding, impact of climate change affecting water quality, microbial communities and GHG-production, we think this indeed is an important driver to include. We have changed the order and first mention figure 3E and then figure 4 in the text.

- Methods - metagenomic analyses: please elaborate on the various parameters and respective version numbers for the various tools and databases used. Indicate also, when the databases were downloaded including version IDs.

These details can be found on github – this would be a long list and would expand the methods section by another half page at least. Parameters are given in the parameters.pl and details on the metagenomic pipelines is provided at <https://github.com/jtamames/SqueezeMeta>. We have also improved the documentation of our code, see <https://github.com/alper1976/chronosequences>

- L 554: describe and highlight the "modifications" used for the C, N, S, marker gene analyses
Details can be found in the R code – listing the KEGGs used as marker for various treats. Still we provided general detail on the modifications. " . . . in particular adding markers for photosynthetic microorganisms and hydrolases (for details see R code in <https://github.com/alper1976/chronosequences>)."

- The code provided should be commented better for reproducibility.
We have improved the documentation of the code by providing READMEs in each code folder and added additional comments to the scripts as well as removed redundant lines in the codes. See <https://github.com/alper1976/chronosequences>